# Delays in obtaining hospital care and abortion-related complications within a context of illegality

Romina M. Hamui[1]*, Estela M. L. Aquino[1], Greice M. S. Menezes[1], Thália Velho Barreto de Araújo[2], Maria Teresa Seabra Soares de Britto e Alves[3], Sandra Valongueiro Alves[2], Maria da Conceição C. Almeida[4]

1 Institute of Collective Health, Federal University of Bahia, Salvador, Bahia, Brazil, 2 Federal University of Pernambuco, Recife, Pernambuco, Brazil, 3 Federal University of Maranhão, São Luís, Maranhão, Brazil, 4 Gonçalo Moniz Institute, Oswaldo Cruz Foundation, Salvador, Bahia, Brazil

* rominahamui@gmail.com

**Data Availability Statement:** All data that corroborate the findings presented are under the responsibility of the coordination of the GravSUS-

## Abstract

Abortion, particularly when illegal, highlights inequities in different populations. Although abortion-related mortality is lower compared to other obstetric causes, abortion complications tend to be more lethal. Delays in seeking and obtaining care are determinants of negative outcomes. This study, nested within the GravSus-NE, analyzed healthcare delays and their association with abortion-related complications in three cities of northeastern Brazil (Salvador, Recife and São Luís). Nineteen public maternity hospitals were involved. All eligible women ≥18 years old hospitalized between August and December 2010 were evaluated. Descriptive, stratified and multivariate analyses were performed. Youden's index was used to determine delay. One model was created with all the women and another with those admitted in good clinical conditions, thus determining complications that occurred during hospitalization and their associated factors. Of 2,371 women, most (62.3%) were ≤30 years old (median 27 years) and 89.6% reported being black or brown-skinned. Most (90.5%) were admitted in good condition, 4.0% in fair condition and 5.5% in poor/very poor condition. Median time between admission and uterine evacuation was 7.9 hours. After a cut-off time of 10 hours, the development of complications increased considerably. Black women and those admitted during nightshifts were more likely to experience a wait time ≥10 hours. Delays were associated with severe complications (OR 1.97; 95%CI: 1.55–2.51), including in the women admitted in good condition (OR 2,56; 95%CI: 1.85–3.55), and even following adjustment for gestational age and reported abortion type (spontaneous/induced). These findings corroborate the literature, highlighting the social vulnerability of women hospitalized within Brazil's public healthcare system in a situation of abortion. The study strongpoints include having objectively measured the time between admission and uterine evacuation and having established a cut-off time defining delay based on conceptual and epidemiological criteria. Further studies should evaluate other settings and new measurement tools for effectively preventing life-threatening complications.

NE study. It is important to emphasize that, as a project approved by the Brazilian National Council of Ethics, the data produced, with sensitive information, have exclusive access and are not openly available. Upon reasonable request and express authorization from the GravSUS-NE study coordination and approval by an ethics committee, controlled access to data for research purpose is possible. If needed, please mail to GravSUS-NE coordinator (estela@ufba.br) (http://www.isc.ufba.br/docentes-e-pesquisadores/), or the Intitute of Colective Health (isc@ufba.br), or the Ethics Committee of the Institute of Collective Health (cepisc@ufba.br).

**Funding:** This study received financial support from the Brazilian Ministry of Health (DECIT) and the Ministry of Science and Technology - National Council for Scientific and Technological Development (CNPq), grants MCT/CNPq/MS-SCTIE-DECIT/CT - Health 22/2007 (#551249/2007-2) and MCT/CNPq/MS/ SCTIEDECIT 54/2008 (#402680/2008-1). In addition, EMLA received a CNPq research productivity grant (306295/2017-2). There was no additional external funding received for this study. URLs to sponsors' websites: https://www.gov.br/saude/pt-br https://www.gov.br/mcti/pt-br https://www.gov.br/cnpq/pt-br The funders had no role in the study design, data collection and analysis, decision to publish, or preparation of the manuscript.

**Competing interests:** The authors have declared that no competing interests exist.

## Introduction

Abortion performed under unsafe conditions constitutes a public health issue that involves potentially avoidable complications [1]. The procedure reflects the inequities present in societies, particularly in settings where its practice is illegal [2, 3]. Although a widely used resource irrespective of social class, race and religious beliefs, access to safe abortion is restricted to a segment of the female population. Indeed, even in settings in which the procedure is illegal, a minority is able to afford an abortion in private clinics or can count on social support networks. For the remaining women, however, abortion trajectories involving unsafe practices result in a risk of health complications and even death [4–7]. Abortion is illegal in Brazil except when performed to save the woman's life or in cases of rape. In 2012, the Brazilian Supreme Court authorized abortion in cases of fetuses with anencephaly.

The introduction of misoprostol in regions with restrictive legislation made clandestine abortions less unsafe. In Brazil, misoprostol is the chosen method in almost half (48%) of all induced abortions [8], successfully reducing, but not eliminating, more aggressive methods that account for more severe complications [9]. This explains a 27% reduction in abortion-related admissions to public hospitals in Brazil between 1995 and 2013 for women in the 10-49-year age group [10]. Nonetheless, a nationwide survey found that 48% of abortions performed in 2016 resulted in complications that required hospitalization [8].

Abortion-related complications are a less common cause of maternal mortality compared to conditions such as hemorrhages and hypertensive disorders; however, once developed, they tend to be more lethal than these obstetric conditions [11]. In Brazil, abortion-related complications represent the fifth most common cause of maternal death [12]; nonetheless, these mortality rates are higher in the northeast of the country [8, 13] and in certain periods [14] abortion-related death rates have surpassed those resulting from other obstetric causes, making it the most common cause of death among younger women [15]. There is evidence that under-reporting of abortion-related complications as a primary cause of death is high compared to the other causes of maternal mortality [16, 17], with values possibly doubling if considered as an associated cause of death [18].

Delays in receiving care have been reported as a key factor in the occurrence of negative obstetric outcomes [3, 19]. The model developed by Thaddeus and Maine [20] identified three moments that define "delays" as determinants of obstetric complications: the first moment is when the symptoms are recognized, and a decision is made to seek help; the second refers to the ability to access an appropriate healthcare service; and the third concerns the woman receiving appropriate and timely treatment at a healthcare unit. This model has been widely used in studies on maternal mortality [21, 22], although to a lesser extent to investigate the complications of abortion.

Postponements both in the decision to seek care [23] and in obtaining appropriate and timely treatment [21, 24] have been highlighted as determining factors in the development of adverse health events. Delays in initiating care have been evaluated as being even more harmful among women with abortion-related complications compared to those with other obstetric conditions; nevertheless, delays between diagnosis and initiating treatment can be up to 20 times greater in the case of abortion [22]. The third delay, vital in understanding the role of healthcare services in the occurrence of negative outcomes, has been the subject of much less investigation [25].

In Brazil, few studies have investigated delays in the provision of care to women in a situation of abortion, particularly with respect to the third delay. A nationwide hospital survey [11] showed that a delay at any time between the woman seeking care and obtaining access to obstetric care was associated with abortion-related complications of increased severity.

Araújo et al. [26] analyzed the first two delays in northeastern Brazil (the multicenter Grav-Sus-NE study), taking into consideration the women's condition at arrival and their peregrination through different healthcare services in search of care. Almost 30% of the women interviewed had experienced difficulties in deciding to seek help and over a quarter of them experienced difficulties in being admitted to hospital. All these delays were associated with more severe morbidities. Also analyzing the GravSus-NE data, Goes investigated barriers to accessing healthcare services, focusing particularly on racial inequalities. One of the findings was that the women who self-reported as black had greater delays in obtaining care, while the principal barrier for seeking care was the fear of being mistreated at the healthcare services [27].

The present study is part of that same original research, delving deeper into these analyses to evaluate the timeliness of post-abortion care in healthcare units, with particular emphasis on the performance of uterine evacuation, as this procedure is central to the nature of the healthcare provided in such circumstances [28]. Uterine evacuation is performed in the majority of cases of abortion [11, 21, 29] and, unless severe complications occur, hospitalization is not required. In fact, women who are better off socioeconomically are able to access this method as outpatients in private clinics even in settings where the procedure is illegal [30].

The objective of this study was to describe the time between admission to hospital and uterine evacuation and to define a cut-off time after which the occurrence of severe complications increases, i.e. the moment that can be defined as a delay in the provision of care. The study also aimed to identify the factors associated with a greater interval of time until the procedure and to investigate the association between delay in performing uterine evacuation and the occurrence of abortion-related complications.

## Materials and methods

### Design and population

This study is nested within the GravSus-NE, a multicenter study conducted in three cities of northeastern Brazil: Salvador, Recife and São Luís. Details of the methodology of GravSus-NE have already been published [31–33]. In brief, it was a hospital-based survey conducted between August and December 2010 and involving all the women of 18 years of age or more admitted to any of the nineteen maternity hospitals in the three aforementioned cities in a situation of abortion or with complications resulting from an abortion, irrespective of the severity of their clinical condition or the type of abortion reported (spontaneous or induced). Cases involving legal pregnancy terminations, ectopic pregnancy, molar pregnancy or any other abnormal product of conception (i.e. all cases for which there are clinical or legal justifications for pregnancy termination under safe conditions) were excluded from the study. Women who had undergone uterine evacuation were considered eligible for the present study irrespective of the procedure used (curettage, manual vacuum aspiration or pharmacological management).

### Data collection

Over a 4-month period, women who were eligible for inclusion in the study were identified daily from among those admitted to hospital. Part of the data was collected at interviews conducted by teams of healthcare professionals specifically trained for the purpose and another part was extracted from hospital records. The interviews, conducted using a structured questionnaire, were performed face-to-face following the woman's discharge from hospital. Next, the hospital records of the women interviewed were identified and a standardized instrument was used to extract data on their clinical condition (condition at arrival and the severity of the

complications presented during hospitalization) and the care provided (date and time of admission and of uterine evacuation).

Conditions on arrival were classified using a consensus conference technique [34] based on a meeting of 25 specialists who were investigators on abortion and/or had experience in intensive care. Matrixes were defined that were used to design data collection forms. The criteria adopted for the classification of severity took into consideration those defined by Pattinson et al. [35], as well as the World Health Organization's definition of near-miss [36], validated in Brazil [37].

Quality control measures included direct field supervision, review of 100% of the interviews by supervisors and review of all the forms on which some type of complication was recorded. To calculate the number of losses, the number of hospitalizations during the period was checked with the hospital admission department. Overall, 2,848 interviews were conducted, with 5.8% losses and 2.7% refusals.

## Data processing and analysis

A database was created using Epi Info for Windows. The analyses were performed using STATA, version 13.0 (StataCorp LP, College Station, TX).

The outcome of interest was the severity of the complications of abortion, initially classified as "non-life-threatening conditions", "potentially life-threatening conditions" and "life-threatening conditions" (including cases of near-miss and death). For the present study, this variable was dichotomized into *severe* and *non-severe complications*, with potentially life-threatening and actually life-threatening conditions being included in the severe category.

The principal independent variable was the time interval between admission and uterine evacuation, with the date and time of these two moments being registered. Youden's index [38] was used to establish a cut-off time after which the occurrence of severe complications increased and to later construct a dichotomous variable regarding whether there had been a delay in receiving care (yes/no). All the analyses took the woman's conditions at admission to hospital into consideration. This was classified based on the matrix produced using the consensus conference of specialists into: good, fair, poor and very poor.

The co-variables taken into consideration included potential effect modifiers and/or confounders: gestational age, type of abortion reported by the woman, education level, self-reported ethnicity/skin color, marital status, number of living children, whether the woman had been admitted during the day or night shift and whether admission occurred on a weekday or weekend.

Descriptive, stratified and multivariate analyses were conducted. The associations were measured using odds ratios (OR). In the stratified analysis, statistical significance was tested using Pearson's chi-square test, with significance level set at $p < 0.05$. The Mantel-Haenszel test was used to analyze confounding and effect modification. A simultaneous analysis of the variables was then conducted using unconditional logistic regression to test the association between a delay in the provision of care and the occurrence of severe complications. The variables selected for adjustment were those with a p-value $< 0.05$ in the saturated models, and the goodness-of-fit was evaluated according to the Akaike (AIC) and Bayesian information criterion (BIC).

Two models were evaluated, one including all the women and the other including only those who arrived at the hospital in good health conditions. The objective here was to evaluate the complications that occurred during the women's stay in hospital.

## Ethical aspects

The internal review boards of the three federal universities responsible for the study approved the study protocol. In respect of the women's autonomy and also to protect them from ever

being identified in the future, a situation that could potentially result in criminal charges, an oral informed consent form was used, which was read to the women and signed only by the investigators. The women were given the right to refuse to answer any of the questions, to refuse to participate or to interrupt the interview at any time. The participating women authorized the investigators to access their hospital records. Procedures were adopted for registering and storing the data that guaranteed anonymity, with the women's names being removed.

## Results

A total of 2,808 women in their first admission to hospital were interviewed: 58.0% in Salvador, 27.1% in São Luís and 13.9% in Recife. Of these, 36 (1.3%) had to be readmitted to hospital during the study; however, data on the second admission do not form part of the present analysis. Hospital records were available for 2,714/2,808 women, with those that could not be found corresponding to 3.3% of the total (0.6% in Salvador, 6.1% in Recife and 7.7% in São Luís). Seventy-six indigenous women (2.8%) and 69 of Asian descent (2.5%) were excluded due to their small numbers and due to the presence of differences that would make it difficult for them to be included in any other group. Twenty-four women were excluded because they had not undergone uterine evacuation during hospitalization. Of the 2,690 who underwent the procedure, data on the date and time of their admission to hospital and/or of uterine evacuation were partially or completely missing from the records in 319 cases, making it impossible to calculate the time interval between these two moments. This resulted in an additional loss of 11.8%, leading to a final study population of 2,371 women.

Comparison of the women who were excluded from the study with those who remained in the analysis showed no statistically significant differences between the two groups with respect to education level, marital status, parity, gestational age, type of abortion reported (spontaneous/induced), or to the women's conditions at arrival at the hospital to which they were admitted. Nevertheless, those excluded from the study tended to be younger (46.4% versus 34.3% were 18 to 24 years old), to be brown-skinned (63.2% versus 52.0%) and to have fewer severe complications (93.4% versus 82.6% had mild complications or none at all) (data not presented).

Most of the women (62.3%) were under 30 years of age (median 27 years) and 89.6% self-reported as being brown- or black-skinned (Table 1). Half the study population had only completed elementary school; 84.8% reported being unmarried at the time of the interview; and only 30.3% did not have children at that time. Gestational age was ≤12 weeks at the time of the abortion in 80.8% of cases, which was reported as having been induced in 26.7% of cases. Regarding the women's admission to hospital, 78.2% were admitted on a weekday and 71.6% during the dayshift.

In 73.4% of cases, the women went directly from their home to the hospital to which they were admitted or (less frequently) from work or from another public space, while 3.5% were transferred from another healthcare unit (Table 1). In general, the women arrived at hospital in good health conditions (90.5%); however, 4.0% arrived in fair condition and 5.5% in poor or very poor condition. Distribution of the women's conditions at admission to hospital did not vary as a function of age, ethnicity/skin color, their place of origin prior to hospitalization or whether they were admitted on a weekday or over the weekend, or on the dayshift or nightshift.

Nevertheless, the women admitted in a poor condition tended to have less education compared to those admitted in a good condition of health (Table 1). The former were more likely to be single, to have had more children, and to be 13 weeks pregnant or more. Indeed, almost twice as many women in this group reported having induced the abortion compared to those

**Table 1. Characteristics of the study population according to health status upon arrival at the hospital.**

| Characteristics | Conditions on arrival at hospital | | | | p-value |
|---|---|---|---|---|---|
| | Good (n = 2,084) % | Fair (n = 92) % | Poor/very poor (n = 127) % | Total (n = 2,303) %[a] | |
| **Age (years)** | | | | | 0.593 |
| 18–24 | 34.1 | 37.0 | 38.6 | 34.4 | |
| 25–29 | 27.7 | 32.6 | 27.6 | 27.9 | |
| 30–34 | 20.9 | 17.4 | 21.2 | 20.8 | |
| ≥35 | 17.3 | 13.0 | 12.6 | 16.9 | |
| **Ethnicity/skin color [b]** | | | | | 0.072 |
| White | 10.9 | 5.4 | 5.6 | 10.4 | |
| Brown | 51.8 | 48.9 | 58.7 | 52.1 | |
| Black | 37.3 | 45.7 | 35.7 | 37.5 | |
| **Education level** | | | | | 0.015 |
| Elementary school | 49.9 | 57.9 | 61.8 | 50.9 | |
| High school/university | 50.1 | 42.1 | 38.2 | 49.1 | |
| **Marital status** | | | | | 0.016 |
| Single | 80.5 | 86.8 | 91.9 | 81.4 | |
| Married/stable union | 15.9 | 9.9 | 6.5 | 15.2 | |
| Widowed/Separated/Divorced [a] | 3.6 | 3.3 | 1.6 | 3.4 | |
| **Number of living children** | | | | | 0.001 |
| 0 | 30.8 | 30.8 | 22.8 | 30.3 | |
| 1 or 2 | 56.0 | 53.8 | 50.4 | 55.6 | |
| ≥ 3 | 13.2 | 15.4 | 26.8 | 14.1 | |
| **Gestational age at admission** | | | | | 0.000 |
| ≤ 12 weeks | 82.6 | 68.2 | 60.5 | 80.8 | |
| ≥ 13 weeks | 17.4 | 31.8 | 39.5 | 19.2 | |
| **Type of abortion reported** | | | | | 0.000 |
| Spontaneous | 75.3 | 51.6 | 55.9 | 73.3 | |
| Induced | 24.7 | 48.4 | 44.1 | 26.7 | |
| **Came directly from:** | | | | | |
| Home/work/public space | 72.9 | 80.4 | 76.2 | 73.4 | 0.109 |
| Healthcare unit (referred or otherwise) | 23.7 | 17.4 | 17.5 | 23.1 | |
| Transferred from another hospital | 3.4 | 2.2 | 6.3 | 3.5 | |
| **Time of admission** | | | | | 0.367 |
| Dayshift | 72.0 | 71.7 | 66.1 | 71.6 | |
| Nightshift | 28.0 | 28.3 | 33.9 | 28.4 | |
| **Day of admission** | | | | | |
| Weekday | 78.4 | 78.3 | 74.8 | 78.2 | 0.642 |
| Weekend | 21.6 | 21.7 | 25.2 | 21.8 | |
| **Severity of abortion complications** | | | | | |
| Mild or none | 90.5 | 14.1 | 4.7 | 82.7 | 0.000 |
| Potentially life-threatening | 9.2 | 85.9 | 91.3 | 16.7 | |
| Near-miss | 0.3 | - | 3.2 | 0.5 | |
| Death | - | - | 0.8 | 0.1 | |

Source: GravSus-NE, Salvador, Recife and São Luís, August–December 2010

[a] Missing data: there were 13 cases in which the woman's conditions at arrival were not registered

[b] Seventy-six indigenous women (2.8%) and 69 women of Asian descent (2.5%) were excluded due to their small numbers and due to the presence of differences that would make it difficult for them to be included in any other group.

**Table 2. Distribution of the time interval (hours) between admission and uterine evacuation according to the occurrence of complications.**

| Interval (hours) | Mild complications or none at all | Severe complications | Total |
|---|---|---|---|
| Number of cases* | 1,908 | 401 | 2,316 |
| 25th percentile (Q1) | 3.4 | 5.3 | 3.6 |
| Median | 7.4 | 11.4 | 7.9 |
| 75th percentile (Q3) | 16.3 | 24.8 | 17.9 |
| Minimum | <0.1 | <0.1 | <0.1 |
| Maximum | 246.8 | 360 | 360 |

* Data regarding the severity of complications were missing in 7 cases.

who were admitted to hospital in good health conditions. Of the women who were admitted in good conditions of health, 9.5% went on to develop severe complications during hospitalization, including 7 of the 11 cases of near-miss.

The median time between admission and undergoing uterine evacuation was 7.9 hours (range <1 to 360 hours). The median time was found to be 54% higher in the cases that went on to develop severe complications and the maximum value was also higher, resulting in a higher interquartile range (Table 2).

Based on Youden's index, a cut-off time of 10 hours was established after which the occurrence of severe complications increased. For this cut-off point, the area under the curve was 0.58, sensitivity 56% and specificity 60% (data not presented). A significant proportion of the women (42.1%) waited 10 hours or more between admission to hospital and undergoing uterine evacuation (Table 3). Black women and those admitted to hospital during nightshifts were more likely to have to wait longer. Overall, 52.2% of the women who had had to wait 10 hours or more before undergoing uterine evacuation considered this time interval to be excessive. There were no statistically significant differences in relation to any of the other sociodemographic characteristics (age, education level, marital status), reproductive characteristics (number of living children, gestational age, type of abortion reported), perception of discrimination, conditions on arrival at the hospital or the day of the week on which they were admitted (weekday or weekend). Most of the women (82.6%) had mild complications or none at all; however, 16.8% had potentially life-threatening complications and there were 11 cases of near miss and one death, totalizing 0.5% of life-threatening conditions during this study (Table 3). Of the women who had had to wait for 10 hours or more between being admitted to hospital and undergoing uterine evacuation, 22.5% presented with severe complications, a 60% increase compared to those for whom this interval was less than 10 hours (13.5%).

In the multivariate logistic regression, when taking all the women into consideration, a delay in performing uterine evacuation was associated with the occurrence of severe complications (OR: 1.97; 95%CI: 1.55–2.51), even following adjustment for gestational age, the type of abortion reported, education level and parity (Table 4). In the subgroup of women in good conditions of health at admission to hospital, a statistically significant association was confirmed between delay and the occurrence of severe complications (OR: 2.56; 95%CI: 1.85–3.55), following adjustment for gestational age and the type of abortion reported.

## Discussion

The women interviewed were mostly young, had few years of formal education, and the majority described themselves as brown-skinned or black. Most were single, but had children. This profile is consistent with the literature and reflects the situation of social vulnerability of women admitted to hospital in a situation of abortion within the Brazilian public healthcare

**Table 3. Characteristics of the study population according to the time interval between admission to hospital and uterine evacuation.**

| Characteristics | Interval between admission and uterine evacuation | | | p-value |
|---|---|---|---|---|
| | <10 hours (n = 1,340) % | ≥10 hours (n = 976) % | Total (n = 2,316) % | |
| **Age (years)** | | | | 0.067 |
| 18–24 | 36.3 | 31.6 | 34.3 | |
| 25–29 | 27.5 | 28.9 | 28.1 | |
| 30–34 | 20.6 | 20.9 | 20.7 | |
| ≥35 | 15.6 | 18.6 | 16.9 | |
| **Ethnicity/skin color [a]** | | | | 0.007 |
| White | 10.6 | 10.0 | 10.4 | |
| Brown | 54.5 | 48.7 | 52 | |
| Black | 34.9 | 41.3 | 37.6 | |
| **Education level** | | | | 0.270 |
| Elementary school | 50.1 | 47.8 | 49.2 | |
| High school/university | 49.9 | 52.2 | 50.8 | |
| **Marital status** | | | | 0.643 |
| Single | 81.6 | 81.2 | 81.4 | |
| Married/stable union | 14.8 | 15.7 | 15.2 | |
| Widowed/Separated/Divorced | 3.6 | 3.1 | 3.4 | |
| **Number of living children** | | | | 0.579 |
| 0 | 31.0 | 29.5 | 30.3 | |
| 1 or 2 | 54.6 | 56.8 | 55.6 | |
| ≥3 | 14.4 | 13.7 | 14.1 | |
| **Gestational Age (weeks)** | | | | 0.081 |
| ≤ 12 | 82.1 | 79.1 | 80.8 | |
| ≥ 13 | 17.9 | 20.9 | 19.2 | |
| **Type of abortion reported** | | | | 0.115 |
| Spontaneous | 72.0 | 75.0 | 73.3 | |
| Induced | 28.0 | 25.0 | 26.7 | |
| **Conditions on arrival at hospital** | | | | 0.212 |
| Good | 91.3 | 89.3 | 90.5 | |
| Fair | 3.4 | 4.8 | 4.0 | |
| Poor/very poor | 5.3 | 5.9 | 5.5 | |
| **Perception of the waiting time until the procedure** | | | | 0.000 |
| Adequate | 68.2 | 47.0 | 59.2 | |
| Too long | 31.2 | 52.2 | 40.1 | |
| Don't know | 0.6 | 0.8 | 0.7 | |
| **Discrimination perceived** | | | | 0.131 |
| No/do not know | 91.8 | 89.9 | 91.0 | |
| Yes | 8.2 | 10.1 | 9.0 | |
| **Time of admission** | | | | 0.000 |
| Dayshift | 76.4 | 65.2 | 71.7 | |
| Nightshift | 23.6 | 34.8 | 28.3 | |
| **Day of admission** | | | | 0.192 |
| Weekday | 77.2 | 79.5 | 78.2 | |
| Weekend | 22.8 | 20.5 | 21.8 | |

(*Continued*)

**Table 3.** (Continued)

| Characteristics | Interval between admission and uterine evacuation | | | p-value |
| --- | --- | --- | --- | --- |
| | <10 hours (n = 1,340) % | ≥10 hours (n = 976) % | Total (n = 2,316) % | |
| **Severity of abortion complications** | | | | 0.000 |
| Mild complications | 86.4 | 77.5 | 82.6 | |
| Potentially life-threatening complications | 13.1 | 22.0 | 16.8 | |
| Near-miss | 0.5 | 0.4 | 0.5 | |
| Death [b] | - | 0.1 | 0.0 | |

Source: GravSus-NE, Salvador, Recife and São Luís, August-December 2010

[a] Seventy-six indigenous women (2.8%) and 69 women of Asian descent (2.5%) were excluded due to their small numbers and due to the presence of differences that would make it difficult for them to be included in any other group

[b] One death occurred.

network. This vulnerability is even clearer among the women admitted to hospital in a more severe condition, with classic risk markers for abortion complications such as higher gestational age at the time of admission and reporting that the abortion was induced [5, 21, 23, 39]. Poor education level has been associated with a greater frequency of abortion in Brazil [8], as well as with abortions at higher gestational ages [40] and greater delays in seeking care [41], increasing the risk of severe complications [39].

In cases of induced abortion, even with the widespread use of misoprostol, women fail to receive accurate information on its use, leading to erroneous administration or use of insufficient doses. In addition, unsafe methods [3] continue to be widely used and can result in poorer conditions at hospital admission. There is evidence that women who induce abortion tend to delay seeking care because they often hesitate before deciding to terminate the pregnancy and are probably afraid of being mistreated and discriminated against at the healthcare services [42]. Furthermore, they are obliged to take more complex trajectories to obtain care [43] and are consequently more likely to develop severe complications [21].

**Table 4. Association of the time interval between admission to hospital and uterine evacuation and the occurrence of severe complications[a].**

| Group evaluated/ Time interval | Saturated Model | | | Adjusted Model | | |
| --- | --- | --- | --- | --- | --- | --- |
| | OR* | 95%CI | p-value | OR | 95%CI | p-value |
| **All the women** | | | | | | |
| ≤10 hours (reference) | 1 | - | - | 1 | - | - |
| >10 hours | 1.97[b] | 1.55–2.51 | 0.000 | 1.96[c] | 1.54–2.48 | 0.000 |
| **Women in good health conditions at admission** | | | | | | |
| ≤10 hours (reference) | 1 | - | - | 1 | - | - |
| >10 hours | 2.56[d] | 1.85–3.55 | 0.000 | 2.49[e] | 1.81–3.42 | 0.000 |

[a] The term "*severe complications*" includes potentially life-threatening conditions, cases of near-miss and death

[b] Adjusted for the woman's age, ethnicity/skin color, education level, marital status, gestational age, reported type of abortion (spontaneous/induced), number of living children, whether admitted to hospital during dayshift or nightshift, and whether admitted to hospital on a weekday or weekend. AIC value: 1788.97. BIC value: 1873.65.

[c] Adjusted for education level, gestational age, reported type of abortion (spontaneous/induced) and number of living children, AIC value: 1786.74. BIC value: 1826.30.[d]

Adjusted for the woman's age, ethnicity/skin color, education level, gestational age, reported type of abortion (spontaneous/induced), number of living children, whether admitted to hospital during dayshift or nightshift, whether admitted to hospital on a weekday or weekend. AIC value: 1133.87. BIC value: 1205.94

[e] Adjusted for gestational age and reported type of abortion (spontaneous/induced). AIC value: 1148.72. BIC value 1171.03.

The conditions of the women arriving at the maternity hospitals are, therefore, affected to a great extent by the first two delays cited in the model established by Thaddeus and Maine [20], as confirmed in earlier GravSus studies. As reported by Araújo [26], the women obliged to take more complex trajectories prior to arriving at the hospital had a poorer health status at admission. These trajectories reflect the barriers faced within the restrictive legal context in Brazil, obliging some women to go from one healthcare unit to another seeking care before they finally succeed in being admitted to hospital. In this setting, black women face greater difficulties in obtaining care, to a great extent due to their fear of being mistreated at the healthcare units [27]. They face greater institutional barriers to being admitted to hospital, suggesting the conjugation of mechanisms involved in racial discrimination and stigma with respect to abortion, particularly where it is illegal.

Even after admission to hospital, one in every five women who had to wait for more than 10 hours for uterine evacuation developed a severe complication as an outcome, with this rate being 60% higher than for the group of women who had to wait less than 10 hours. Two factors were associated with this greater interval: being admitted to hospital during a nightshift and self-reporting their ethnicity as black.

Cases arriving at emergency departments at night and in the early hours of the morning tend to be more severe and this may reflect a need for additional procedures prior to uterine evacuation. Furthermore, reasons associated with the hierarchy of priorities may perhaps explain the longer wait for those women admitted at night, when there are fewer staff members at the hospitals, with cases considered "legitimate" (i.e. women giving birth) possibly being given priority [42, 44]. The international literature shows that both the diagnostic conclusion and the time until interventions are performed are affected by a so-called "implicit bias" in healthcare professionals with respect to the women [45]. This bias refers to racial inequalities [46] but also affects those women with stigmatizing conditions such as abortion in Brazil.

The median time interval of 7.9 hours between admission to hospital and undergoing uterine evacuation was consistent with data from earlier studies in which values range from 6 [47] to 24 hours [48]. Part of this time can be explained as being the result of technical issues related to the clinical management and adequate preparation for a procedure that is often surgical (curettage or manual vacuum aspiration). Indeed, surgical interventions may require fasting and use of antibiotics, as well as misoprostol for cervical ripening [49]. These intervals of time can vary according to the woman's clinical conditions. Women arriving at the hospital in severe conditions require faster measures to be taken right from the time of their admission [50]. Nevertheless, when conditions at arrival are good, there is still no clinical reason for a delay of more than nine or ten hours, particularly since after this time the occurrence of complications is greater, as shown in the present study. The distribution of this interval as a continuous variable showed greater median and maximum values in those cases in which severe complications developed, confirming the association between delays and the occurrence of these complications.

Receiving care promptly at admission, however, does not guarantee adequate continuity of care. In a survey conducted in 2012 with women admitted to maternity hospitals in southeastern Brazil in a situation of abortion, Adesse et al. [50] found that, although the nursing staff had performed risk classification of the women upon their arrival at the hospital, subsequent care was not provided within the recommended time. Therefore, data on the hospital records showed that, although 90% of the women had been classified for risk within 30 minutes of their arrival at the emergency department, this did not mean that those at greater risk were referred for immediate medical evaluation or admission to hospital, as recommended in the classification.

The present study shows that the women who had to wait longest for uterine evacuation (more than 10 hours) were more likely, compared to those who waited for less than 10 hours, to consider the delay excessive. Aquino et al. [31] also used data from the GravSus-Ne study and showed that most of the women evaluated the time they had to wait and the treatment they received as being good. Nevertheless, those results must be interpreted with caution within contexts such as Brazil where access to healthcare units in a situation of abortion can be difficult, since relief at finding that their physical health has been restored can lead the women to give a positive evaluation of the care received, even if the minimum quality established in the technical guidelines was not reached.

Not all the factors associated with a greater delay could be analyzed in the present study. Technical requirements (such as the need to wait for the cervix to ripen prior to evacuation in cases of retained products of conception) and problems related to the organization of healthcare services (such as the availability of equipment and trained staff to perform ultrasonography on a 24-hour basis) were not evaluated. Nevertheless, previous analyses using data from the GravSus-Ne study identified institutional barriers that suggested delays in providing hospital care [26, 27]. Góes et al. [27] showed how women who identified as black, when compared to brown-skinned or white women, were more likely to report institutional difficulties in being admitted to hospital, both due to a lack of beds and to having to wait for women in labor who were given priority. These delays were not associated with better clinical conditions upon arrival at the hospital, since conditions were found to be more severe in black women at admission compared to white women. However, it cannot be ruled out that situations of racial discrimination, indicating practices of institutional racism, could have been reproduced following admission, with the women already in hospital, and could have resulted in delays in receiving care.

Another factor is discrimination in providing care due to professionals taking a moral stance under the presumption that the abortion was induced. Women suffering a miscarriage can initially be subject to discrimination, suspected of having induced an abortion until proven otherwise [44]. Delaying the provision of care as a veiled punishment for the act, reported particularly by women who had an induced abortion, is present together with other expressions of discrimination such as lack of consent to procedures, violation of privacy and confidentiality, and even verbal abuse [42, 51].

Most of the women who arrived in good health conditions, without having been to another healthcare unit prior to being admitted to hospital and who were only requesting to complete uterine evacuation, had no prior complications. However, 9.5% of these women developed complications during hospitalization. Under these circumstances, the delay in performing the procedure was found to be associated with the occurrence of severe complications (OR: 2.56; 95%CI: 1.85–3.55), even following adjustment for gestational age and the type of abortion reported, thus highlighting the relevance of the proposed analysis on the third delay, in consonance with the international literature. Studies on this subject are often based on medical audits and reviewing patient records, evaluated subjectively by specialists without objectively measuring the duration of the delay [19, 52]. One of the few exceptions is the study conducted by Melese et al. [21] in which the delay is stipulated as an interval of 6 hours; however, there is no explanation regarding the criteria used.

Some methodological aspects merit discussion. The population of the present study consists entirely of women living in urban areas who are clients of the public healthcare system. Women under 18 years of age, who confront greater difficulties in accessing healthcare services, with a consequently greater risk of developing abortion-related complications, were not included [53].

Women transferred from another healthcare unit (3.5%) and women who had to be re-admitted to hospital (1.3%) were not included in the present analysis since this would demand special procedures for data collection and analysis. In both cases, however, their inclusion would probably add to the association found, since the time until definitive resolution of the complications would necessarily consider the entire duration of the different hospitalizations.

The strongpoint of the present study was the inclusion of all the women admitted to hospital for abortion-related complications in the three cities investigated through the use of a hospital census, assuring that all the cases with more severe complications, particularly those who survived (near-miss) were included. Although some hospital records could not be located at all, the characteristics of the cases excluded for different reasons were comparable to those included in the analysis. Taken together, both *records not located* and *patients excluded*, these cases were few and unlikely to change the present conclusions substantially.

The originality of the present study lies in the objective measurement of the time interval and the establishment of a cut-off time defining the third delay, supported by conceptual and epidemiological criteria in accordance with the model proposed by Thaddeus and Maine [20]. Uterine evacuation was chosen as the indicator of the timeliness of care, since this is a procedure considered by the World Health Organization as the definitive treatment for abortion-related complications [54] and is performed in the great majority of cases of women arriving at hospital without severe complications.

By establishing the relevance of the third delay in the occurrence of abortion-related complications in the women who arrived at hospital in good health conditions, including seven of the eleven cases of near-miss, the present study fills a gap in the knowledge available in the literature on this subject, which has historically given priority to the obstetric complications of full-term pregnancies, particularly emphasizing cases involving deaths.

## Conclusions

This study evaluated public healthcare services in Brazil, where abortion laws are restrictive, and highlighted the role of the third delay within the provision of abortion care in the occurrence of severe complications. Nevertheless, further studies on this delay are required in settings with similar legislation. Likewise, new measurement tools need to be developed to demonstrate the effects of these events on women's lives, thus contributing to the implementation of actions that would guarantee full respect of women's sexual and reproductive rights.

Even in settings similar to that of Brazil where legislation is restrictive and there is a strong stigma with respect to abortion, rethinking the care model is possible. Care should be guaranteed in spaces other than maternity hospitals, using the day-hospital format, for cases in which this is possible, reserving post-abortion hospital care for more severe cases in physical spaces that are separate from those reserved for women giving birth. For the great majority of women in a situation of abortion, uterine evacuation should be performed using less invasive techniques such as a medical abortion or manual vacuum aspiration, which can be performed under local anesthesia, as suggested by the World Health Organization and recognized in Brazilian guidelines [55].

Abortion continues to be illegal in the country, with the only change being the incorporation by the Supreme Court in 2012 of a third legal exception to the law–in cases of anencephaly. Therefore, no change has occurred in unsafe abortion care within the public health network.

With the Covid-19 pandemic, reproductive healthcare has been strongly affected worldwide, with a reduction in access to contraception and to abortion [56]. In settings such as Brazil and other Latin American countries where access to these services is already difficult due to

restrictive legislation, the pandemic compounded these difficulties. Abortion care, which is considered an elective procedure even in cases in which the procedure is legal, was relegated to the background in order to prioritize measures to combat Covid-19 [57]. Therefore, it has become even more urgent to review the current protocols of abortion care under these circumstances, including the implementation of new technologies such as telemedicine, which is known to be effective and safe [58–60]. All these changes would encourage respect for women's health and life by offering humanized treatment of abortion.

## Acknowledgments

The authors are grateful to the women participating in this study, who generously shared their stories.

## Author Contributions

**Conceptualization:** Romina M. Hamui, Estela M. L. Aquino, Thália Velho Barreto de Araújo, Maria Teresa Seabra Soares de Britto e Alves, Sandra Valongueiro Alves.

**Data curation:** Romina M. Hamui.

**Formal analysis:** Romina M. Hamui, Estela M. L. Aquino, Greice M. S. Menezes, Maria da Conceição C. Almeida.

**Funding acquisition:** Estela M. L. Aquino.

**Investigation:** Romina M. Hamui, Thália Velho Barreto de Araújo, Maria Teresa Seabra Soares de Britto e Alves, Sandra Valongueiro Alves.

**Methodology:** Romina M. Hamui.

**Project administration:** Estela M. L. Aquino.

**Resources:** Estela M. L. Aquino.

**Visualization:** Romina M. Hamui.

**Writing – original draft:** Romina M. Hamui, Estela M. L. Aquino, Greice M. S. Menezes, Maria da Conceição C. Almeida.

**Writing – review & editing:** Romina M. Hamui, Estela M. L. Aquino, Greice M. S. Menezes, Thália Velho Barreto de Araújo, Maria Teresa Seabra Soares de Britto e Alves, Sandra Valongueiro Alves, Maria da Conceição C. Almeida.

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
