## [Decision Letter · Decision Letter 0]

3 Mar 2023

PONE-D-22-23361Delays in obtaining hospital care and abortion-related complications within a context of illegalityPLOS ONE

Dear Dr. Romina,

Thank you for submitting your manuscript to PLOS ONE. After careful consideration, we feel that it has merit but does not fully meet PLOS ONE’s publication criteria as it currently stands. Therefore, we invite you to submit a revised version of the manuscript that addresses the points raised during the review process.

We look forward to receiving your revised manuscript.

Kind regards,

Surangi Nilanka Jayakody Mudiyanselage, MBBS, MD

Academic Editor

PLOS ONE

Journal Requirements:

“EMLA

This study received financial support from the Brazilian Ministry of Health (DECIT) and the Ministry of Science and Technology - National Council for Scientific and Technological Development (CNPq), grants MCT/CNPq/MS-SCTIE-DECIT/CT - Health 22/2007 (#551249/2007-2) and MCT/CNPq/MS/ SCTIEDECIT 54/2008 (#402680/2008-1). Additionally, a CNPq research productivity grant (306295/2017-2) was provided.

https://www.gov.br/saude/pt-br

https://www.gov.br/mcti/pt-br

https://www.gov.br/cnpq/pt-br

Reviewers' comments:

Reviewer's Responses to Questions

**Comments to the Author**

1. Is the manuscript technically sound, and do the data support the conclusions?

Reviewer #1: Yes

Reviewer #2: Yes

2. Has the statistical analysis been performed appropriately and rigorously? 

Reviewer #1: Yes

Reviewer #2: Yes

3. Have the authors made all data underlying the findings in their manuscript fully available?

Reviewer #1: No

Reviewer #2: No

4. Is the manuscript presented in an intelligible fashion and written in standard English?

Reviewer #1: Yes

Reviewer #2: Yes

5. Review Comments to the Author

Reviewer #1: This is a further analysis of a study conducted in 2010, it contributes to the understanding of the delay in performing uterine evacuation and occurrence of abortion-related complications.

The author has provided reasonable explanation regarding the restrictions in availability of data in the manuscript (review question (3)).

Comments:

1. Page 5 - Introduction

I suggest including information regarding the legal status of abortion in the areas where the study was conducted as it is not clear. There is a statement on regions with restrictive legislation in line 74, however, it is not clear whether this include the regions of study. The meaning of restrictive legislation for abortion needs to be explained.

2. Page 9, line 178:

The statement ‘to calculate the number of losses’ is not clear; does ‘number of losses’ means ‘missing data’? This need clarification.

3. Tables 1 and 3:

To state the exact number of indigenous women and those of Asian descent which were excluded.

Reviewer #2: The authors of this manuscript have described " Delays in obtaining hospital care and abortion-related complications within a context of illegality" technically sound manner.

However, the original study was conducted in 2010 which is more than 12 years back. Publishing an article and applicability of the recommendations based on a study which was conducted more than 12 years back is arguable.

Table 2 is on time interval between admission and uterine evacuation according to the occurrence of complications. If the data are being skewed, better explain with median and IQR rather than explaining all mean, SD and median. Also author can mention the number of cases outside from this table.

Table 3 - If Asian and Indigenous women were excluded, under ethnicity author can mention the number as n = .......

The authors have included the recommendations also under conclusion.

Better if the limitations have been mentioned.

6. PLOS authors have the option to publish the peer review history of their article (what does this mean?). If published, this will include your full peer review and any attached files.

Reviewer #1: No

Reviewer #2: No

---

## [Author Response · Author response to Decision Letter 0]

30 Apr 2023

In response to the academic editor’s suggestions:

1. Please ensure that your manuscript meets PLOS ONE style requirements, including

those for file naming. The PLOS ONE style templates can be found at

https://journals.plos.org/plosone/s/file?id=wjVg/PLOSOne_formatting_sample_main_b

ody.pdf and

https://journals.plos.org/plosone/s/file?id=ba62/PLOSOne_formatting_sample_title_aut

hors_affiliations.pdf.

The text has been reviewed once again to ensure that we meet with your requirements and to the best of our knowledge we believe that we have complied with the PlosOne guidelines

Please provide an amended statement that declares *all* the funding or sources of support (whether external or internal to your organization) received during this study, as detailed online in our guide for authors at http://journals.plos.org/plosone/s/submit-now. 

Please also include the statement “There was no additional external funding received for this study.” in your updated Funding Statement.

An amended statement has now been included in the cover letter, as requested:

Funding statement: 

This study received financial support from the Brazilian Ministry of Health (DECIT) and the Ministry of Science and Technology - National Council for Scientific and Technological Development (CNPq), grants MCT/CNPq/MS-SCTIE-DECIT/CT - Health 22/2007 (#551249/2007-2) and MCT/CNPq/MS/ SCTIEDECIT 54/2008 (#402680/2008-1). In addition, EMLA received a CNPq research productivity grant (306295/2017-2). There was no additional external funding received for this study.

URLs to sponsors’ websites:

https://www.gov.br/saude/pt-br

https://www.gov.br/mcti/pt-br

https://www.gov.br/cnpq/pt-br

The funders had no role in the study design, data collection and analysis, decision to publish, or preparation of the manuscript.

The reference list has been checked as requested. Two of the references have been updated:

6 - Shah I, Ahman E. Unsafe abortion in 2008: global and regional levels and trends. Reprod Health Matters. 2010;18:90-101. https://doi.org/10.1016/S0968-8080(10)36537-2.

27 - Goes EF, Menezes GMS, Almeida MC, Araújo TVB, Alves SV, Alves MTSSBE, et al. Racial vulnerability and individual barriers for Brazilian women seeking first care following abortion. Cad Saude Publica. 2020;36(Suppl 1):e00189618. https://doi.org/10.1590/0102-311X00189618. 

In response to Reviewer 1:

1. Page 5 - Introduction

I suggest including information regarding the legal status of abortion in the areas where the study was conducted as it is not clear. There is a statement on regions with restrictive legislation in line 74, however, it is not clear whether this includes the regions of study. The meaning of restrictive legislation for abortion needs to be explained.

We are grateful to the reviewer for this suggestion. Text has now been added to the Introduction, as follows (Page 5, lines 72-74):

Abortion is illegal in Brazil except when performed to save the woman's life or in cases of rape. In 2012, the Brazilian Supreme Court authorized abortion in cases of fetuses with anencephaly.

2. Page 9, line 178:

The statement ‘to calculate the number of losses’ is not clear; does ‘number of losses’ means ‘missing data’? This need clarification.

We are particularly grateful for this comment as it gave us the opportunity to correct and provide a clearer explanation regarding these losses. To identify those eligible for the survey, the women admitted daily to each of the hospitals involved in the study were checked and a list of those to be interviewed by the investigators was prepared. A total of 2,848 interviews were carried out, with 5.8% of losses and 2.7% of refusals. This information has now been added to the text. (Page 9, lines 184-185)

3. Tables 1 and 3:

To state the exact number of indigenous women and those of Asian descent, which were excluded.

Thank you for this suggestion. The information requested has now been added to page 12, lines 243-245 and as a footnote to both tables, as follows:

Seventy-six indigenous women (2.8%) and 69 women of Asian descent (2.5%) were excluded due to their small numbers and due to the presence of differences that would make it difficult for them to be included in any other group.

In response to Reviewer Nº2:

1) The authors of this manuscript have described " Delays in obtaining hospital care and abortion-related complications within a context of illegality" technically sound manner. However, the original study was conducted in 2010 which is more than 12 years back. Publishing an article and applicability of the recommendations based on a study which was conducted more than 12 years back is arguable.

Thank you for this comment. The following text has now been added to the Conclusion:

Abortion continues to be illegal in the country, with the only change being the incorporation by the Supreme Court in 2012 of a third legal exception to the law – in cases of anencephaly. Therefore, no change has occurred in unsafe abortion care within the public health network.

2) Table 2 is on time interval between admission and uterine evacuation according to the occurrence of complications. If the data are being skewed, better explain with median and IQR rather than explaining all mean, SD and median. Also author can mention the number of cases outside from this table

Thank you for pointing this out. The Shapiro-Wilk test was performed and the distribution of the data is indeed not normal. Therefore, results have now been reported as medians. Means and SD have been removed from the table, as suggested. (Page 16).

3) Table 3 - If Asian and Indigenous women were excluded, under ethnicity author can mention the number as n = .......

Thank you for this suggestion. This information has now been added to the Results section (page 12, lines 243-245) and as a footnote to Tables 1 and 3, as follows:

Seventy-six indigenous women (2.8%) and 69 women of Asian descent (2.5%) were excluded due to their small numbers and due to the presence of differences that would make it difficult for them to be included in any other group.

4) The authors have included the recommendations also under conclusion. Better if the limitations have been mentioned.

We chose to present the limitations and strengths of the study in the Discussion (lines 479-498), reserving the summary of the main findings and their implications for future research and the reorganization of abortion care for the Conclusions. We would prefer to retain the manuscript in this format; however, should the editor deem it necessary, we are willing to make the changes suggested by the

---

## [Editor Report · Decision Letter 1]

30 May 2023

Delays in obtaining hospital care and abortion-related complications within a context of illegality

PONE-D-22-23361R1

Dear Dr. Hamui,

We’re pleased to inform you that your manuscript has been judged scientifically suitable for publication and will be formally accepted for publication once it meets all outstanding technical requirements.

Kind regards,

Surangi Jayakody, MBBS, MSc, MD

Academic Editor

PLOS ONE

---

## [Editor Report · Acceptance letter]

7 Jun 2023

PONE-D-22-23361R1 

Delays in obtaining hospital care and abortion-related complications within a context of illegality 

Dear Dr. Hamui:

I'm pleased to inform you that your manuscript has been deemed suitable for publication in PLOS ONE. Congratulations! Your manuscript is now with our production department. 

Kind regards, 

on behalf of

Dr Surangi Jayakody 

Academic Editor

PLOS ONE